# Platelets, Bacterial Adhesins and the Pneumococcus

**DOI:** 10.3390/cells11071121

**Published:** 2022-03-25

**Authors:** Kristin Jahn, Thomas P. Kohler, Lena-Sophie Swiatek, Sergej Wiebe, Sven Hammerschmidt

**Affiliations:** Center for Functional Genomics of Microbes, Department of Molecular Genetics and Infection Biology, Interfaculty Institute for Genetics and Functional Genomics, University of Greifswald, 17489 Greifswald, Germany; kristin.jahn@uni-greifswald.de (K.J.); thomas.kohler@uni-greifswald.de (T.P.K.); lena-sophie.swiatek@uni-greifswald.de (L.-S.S.); sergej.wiebe@med.uni-greifswald.de (S.W.)

**Keywords:** *Streptococcus pneumoniae*, platelet killing, platelet activation, pore formation, surface proteins, toxin, pneumolysin, *Staphylococcus aureus*, MSCRAMMs

## Abstract

Systemic infections with pathogenic or facultative pathogenic bacteria are associated with activation and aggregation of platelets leading to thrombocytopenia and activation of the clotting system. Bacterial proteins leading to platelet activation and aggregation have been identified, and while platelet receptors are recognized, induced signal transduction cascades are still often unknown. In addition to proteinaceous adhesins, pathogenic bacteria such as *Staphylococcus aureus* and *Streptococcus pneumoniae* also produce toxins such as pneumolysin and alpha-hemolysin. They bind to cellular receptors or form pores, which can result in disturbance of physiological functions of platelets. Here, we discuss the bacteria-platelet interplay in the context of adhesin–receptor interactions and platelet-activating bacterial proteins, with a main emphasis on *S. aureus* and *S. pneumoniae*. More importantly, we summarize recent findings of how *S. aureus* toxins and the pore-forming toxin pneumolysin of *S. pneumoniae* interfere with platelet function. Finally, the relevance of platelet dysfunction due to killing by toxins and potential treatment interventions protecting platelets against cell death are summarized.

## 1. Introduction

Platelets are anucleated, discoid shaped cells of the blood with a size of 2–4 µm in diameter [1]. Platelets are derived from megakaryocyte (MK) shedding in the bone marrow and the lungs [2,3]. Hence, their translation repertoire is limited to stable MK-derived mRNA [4]. Important platelet functions contribute to coagulation and closure of vascular damage occurring during, e.g., microbial infections. The main players during this processes are the surface expressed group of platelet glycoproteins. They are expressed in high numbers such as, e.g., integrin αIIbβ3, which is highly abundant on the surfaces of platelets and megakaryocytes and is involved in the crosslinking/aggregation of single platelets after activation [5]. Integrin αIIbβ3 is the fibrinogen receptor but also interacts with other extracellular matrix (ECM) proteins containing an RGD-like motif such as von-Willebrand-Factor (vWF), thrombospondin-1 (TSP-1), or fibronectin [6]. Platelet factors playing a role in hemostasis and infection are stored in different cytoplasmic granules. These granules undergo exocytosis upon platelet stimulation/activation and release their content into the circulation or granule proteins re-associate to the platelet surface [7]. Three types of granules can be distinguished: alpha-granules, dense granules, and lysosomal granules. Dense granules contain adenosine diphosphate (ADP), adenosine triphosphate (ATP), serotonin, histamine, and ions such as Ca^2+^. Alpha granules contain platelet factor 4 (PF-4, CXCL4), P-selectin, coagulation factors such as factor V, mitogenic factors, adhesive glycoproteins such as vWF, TSP-1, and fibrinogen, but also microbicidal proteins. Glycosidases, which are important for clot retraction, are mainly stored in the lysosomal granules [7].

However, in addition to their role in hemostasis, platelets have other important functions. In recent years, platelets have been increasingly recognized as immune and inflammatory cells. Several studies have highlighted the role of platelets in acute and chronic inflammatory processes such as stroke, myocardial infarction, infections, and sepsis [8,9,10,11,12]. Indeed, platelets are the most abundant circulating cell type with important immune functions. They display their function as immune cells either locally at sites of platelet activation or systemically by interacting with, e.g., leukocytes or via release of immune modulatory molecules [13,14]. Platelets store more than 300 proteins in their granules [15]. Besides the abovementioned proteins, granules also contain proteins acting as chemokines and cytokines such as, e.g., RANTES (Regulated upon Activation, Normal T Cell Expressed and Presumably Secreted) or CXCL12 (stromal cell-derived factor 1), and recruit and stimulate other cells of the immune system or induce endothelial inflammation [16]. Granule release also leads to changes in the platelet membrane composition. An increased number of integrin αIIbβ3 molecules and increased surface expression of P-selectin (CD62P) can be found on the surface after platelet activation. Exposure of P-selectin mediates initial interactions between platelets and leukocytes via immobilization of leukocytes at the site of lesion [17] and is used as a binding partner for initiation of complement activation [18]. In addition, platelets are a peripheral source of serotonin, stored in dense granules, which leads to differentiation of monocytes into dendritic cells (DCs) and also T-cell activation [19].

## 2. Platelets as Immune Cells in Infections

The first systemic response of the body to any kind of infection, tissue injury, or trauma is known as the acute phase response (APR). During APR, proinflammatory cytokines are released and acute phase proteins are produced [20]. By impairing microbial growth and promoting procoagulant activity to trap pathogens in local blood clots, platelets play a crucial role in the APR [21]. In addition, next to inflammatory and immune cells, platelets also produce interleukin 1-β (IL-1β) [21], which is not stored in granules as ready-to-use protein, but instead is translated from megakaryocyte (MK)-derived mRNA and released upon stimulation [22]. In mouse models of malaria, IL-1β has been demonstrated to play a major role in induction of the APR [21]. Another aspect that makes platelets part of the innate immune system is the expression of pattern recognition receptors (PRR) such as toll-like receptors (TLRs). TLR4, for example, is able to recognize bacterial lipopolysaccharides (LPS), which leads to platelet activation and release of IL-1β-rich microparticles, promoting interaction and activation with, e.g., endothelial cells [23]. However, others did not observe activation of washed platelets after incubation with LPS. Only incubation of whole blood with LPS led to increased P-selectin levels [24], indicating an indirect effect of LPS on platelets. During bacterial infections, TLR2 can be stimulated, leading to formation of platelet/neutrophil aggregates, which enhances adhesion of the aggregates to sites of injury or infection [25]. In addition, TLR2 stimulation of MKs is followed by increased maturation of MKs and elevated protein content, suggesting effects on platelet count, platelet function, and inflammation [26].

Besides their role in innate immunity, platelets are also pivotal for the acquired immune response. Platelets stimulate T-cell activation, trafficking, and also differentiation [27]. T cells are divided, with MHC class II recognizing CD4^+^ cells and MHC class I recognizing cytotoxic CD8^+^ cells. CD4^+^ cells release cytokines regulating the activity of B cells and innate immune cells and can be divided into immune effector cells (Th1, Th2, Th17) and T-regulatory cells. Platelet-derived chemokines such as RANTES trigger activation and arrest of T cells at sites of infections. Upon release, RANTES is immobilized on the endothelium and triggers the arrest of monocytes and monocyte-derived cells at the endothelium but not on adherent platelets in a shear resistant manner (Figure 1) [28]. Adhesion of rolling monocytes on activated platelets as well as on the exposed endothelium is P-selectin-dependent, and this interaction further triggers expression and secretion of cytokines such as TNF-α [29,30,31]. Furthermore, CXCL4 (PF-4) can mediate T-cell trafficking to sites of injury or infection by direct and indirect stimulation of CXCR3 expression on T cells [14]. In addition, T cells are also able to activate platelets via CD40L/CD40 interaction, leading to the release of chemokines such as RANTES and an increase in T cell recruitment [32,33]. Platelets further recruit and activate DCs via CD11b-JAM-C interactions, and the increased expression of CD80 and CD86 leads to enhanced T-cell response [34,35].

Another immune cell function of platelets is displayed by the release of microvesicles. The so-called platelet microvesicles (PMVs) are lipid membrane vesicles with a size of 0.1–1.0 µm that are mediators of cell–cell communication. Elevated numbers of PMVs in the circulation are associated with inflammation and diseases such as arthritis [36], development of an acute coronary artery syndrome, and stroke [37,38]. PMVs carry adhesion molecules (CD62P, RANTES) that facilitate monocyte arrest at sites of deposited PMVs (vessel walls) and additional recruitment of activated platelets [39]. PMVs contain up to 250 different microRNAs (miRNAs, 20–22 nucleotides long). These miRNAs and also other stored molecules can be transferred to other cell types such as immune cells, vascular cells, and also smooth muscle cells, thereby changing their gene expression [40,41]. Changes in gene expression include phenotypic switches of monocytes and monocyte-derived cell lines such as macrophages towards a phagocytic phenotype (Figure 1) [42]. Furthermore, not only PMVs released upon platelet activation but also upon apoptosis of platelets have immunomodulatory effects, as shown by differentiation of monocytes into phagocytes [43]. In bacterial infections, platelets can contribute to clearance of bacteria via release of antimicrobial peptides present in PMVs or stimulation of immune cells via release of immunomodulatory molecules. PMVs can be distinguished in kinocidins, defensins, thymosins, and derivatives of antimicrobial peptides, which act against *Staphylococcus* aureus (*S. aureus*) and *Candida albicans* [44,45].

## 3. Interactions of Platelets with Bacteria

Bacteria are able to spread from the site of infection, thereby often crossing host barriers and entering the circulatory system, leading to bacteremia and sepsis. Complications of bacteremia associated with abnormal platelet functions are, e.g., infective endocarditis and disseminated intravascular coagulation (DIC) [46]. Interactions between platelets and bacteria are characterized by direct (Figure 2) or indirect binding of bacteria to platelets or via released bacterial factors [24,46,47]. Indirect binding occurs via bridging molecules of the extracellular matrix (ECM), thereby linking bacterial surface proteins with platelet receptors [48]. The adhesive properties of bacteria towards platelets are essential for colonization of, e.g., cardiac valves during infective endocarditis [49,50]. Colonization can lead to local and/or systemic infections, and this might result in platelet activation in the bloodstream. Some bacterial infections cause severe thrombocytopenia without preceding bacteremia. To these belong, e.g., *Helicobacter pylori*-mediated immune thrombocytopenia driven by autoantibody-destroying platelets [51,52] or the hemolytic uremic syndrome (HUS) caused by Shiga toxin-producing *Escherichia coli* (*E. coli*) [52,53], which is due to amplification of the platelet activation process. Thrombocytopenia associated with bacterial infections appears as a secondary effect of systemic platelet activation, coagulation, and DIC, due to boosted platelet consumption [54].

Besides platelet activation as a result of bacterial binding, another mode of bacteria–platelet interaction has been discovered, namely, internalization of bacteria by platelets. *S. aureus* was the first bacterium that was described to be internalized by platelets [55,56]. A prerequisite for internalization of *S. aureus* is a simultaneous platelet stimulation by ADP. Slightly differently, *Porphyromonas gingivales* was shown to be internalized by platelets without additional stimulation of platelets [57,58]. Furthermore, the platelet FcγRII receptor might also be able to initiate internalization of IgG–bacteria complexes as it has been shown for beads (0.5–1.5 µm in size) coupled with IgG or *E. coli* pre-opsonized with IgGs [59,60]. Other studies demonstrated a kind of searching and shuttling of invading bacteria by platelets. Adherent platelets can migrate over their substrate, collecting all substrate-bound material including bacteria, resulting in boosted activity of phagocytes [61]. In addition, platelets were shown to deliver the intracellular bacterium *Listeria monocytogenes* to dendritic cells [62]. However, the fate of the internalized bacteria is still unclear. On the one hand, they could be killed by antimicrobial substances of α-granules. On the other hand, the intracellular fate might help the bacteria to escape from the host immune system.

## 4. Platelet Receptors in Bacterial Infections and Bacterial Adhesins

Platelets express a large number of receptors involved in interactions with pathogens. This includes integrins, G-protein-coupled receptors, ADP receptors (purinergic receptors, P2Y), leucine-rich repeat glycoproteins (GP), Toll-like receptors (TLRs), IgG superfamily receptors (GPVI, FcγRIIa), and tyrosine kinase receptors [63]. The FcγRIIa receptor is a low-affinity IgG receptor binding to the Fc part of immunogloblins [64]. About 2000 to 3000 FcγRIIa receptors are expressed on the surface of a single platelet [65], which enable binding and internalization of immune complexes containing IgGs (Figure 3) [59]. Binding of, e.g., IgG-covered pathogens leads to platelet activation and aggregation. Binding and activation of other platelet receptors by bacteria often requires simultaneous stimulation of FcγRIIa. For example, platelet aggregation induced by *E. coli* is dependent on simultaneous stimulation of integrin αIIbβ3 and FcγRIIa [66]. For *S. aureus* and *S. epidermidis*, a clustering of integrin αIIbβ3 or TLR with FcγRIIa is necessary for platelet activation [67]. Some pathogens such as *S. aureus* or *H. pylori* require plasma proteins such as fibrinogen or vWF to crosslink FcγRIIa with GPIb receptors for platelet activation [68].

As mentioned above, integrin αIIbβ3 interacts with ECM proteins containing an RGD-like motif [6]. Some bacteria express surface proteins with domains rich in serine-aspartate repeats. This is highlighted by a family of *S. aureus* surface components called microbial surface components recognizing adhesive matrix molecules (MSCRAMMs). They exhibit sequence repeats mediating adherence to platelets or other host cell tissues as one of the first steps during infection [69]. Well-characterized members of this family are clumping factor A and B (ClfA and ClfB), fibronectin-binding protein A and B (FnBPA and FnBPB), and serine-aspartate repeat-containing protein E (SdrE). Fibrinogen bridges ClfA and ClfB via their fibrinogen binding domains to integrin αIIbβ3 [70], whereas fibrinogen or fibronectin is used by FnBPA and FnBPB for bridging [71,72]. In addition, integrin αIIbβ3 is also directly targeted by ClfA, but not ClfB, resulting in platelet activation (Figure 3) [73,74]. Further, integrin αIIβ3 can be directly bound by the *S. aureus* proteins iron-regulated surface determinant B (IsdB) [75]. Next to pneumococci, fibrinogen also bridges proteins of other bacteria such as serine-aspartate dipeptide repeat G (SdrG) protein of *S. epidermidis* to integrin αIIbβ3 [76]. A further platelet protein target of bacteria is the surface-associated protein-disulfide polymerase (PDI). The *S. aureus* extracellular adherence protein (Eap), a member of the SERAM (secretable expanded repertoire adhesive molecules) family, binds directly but also indirectly utilizes fibrinogen as a bridging molecule to PDI, resulting in platelet activation [77]. Direct activation and aggregation of platelets is also achieved by other secreted *S. aureus* proteins such as the chemotaxis inhibitory protein (CHIPS), the formyl peptide receptor-like 1 inhibitory protein (FLIPr), and the major autolysin (AtlA) (Figure 3) [24]. Upon activation of platelets, the extracellular fibrinogen-binding protein EfB of *S. aureus* binds to surface-exposed P-selectin and inhibits interactions between platelets and leukocytes [78,79].

GPIbα is a type-1 glycosylated membrane receptor that is highly abundant on the surface of platelets and megakaryocytes [80]. GPIbα is found in a complex with GPIbβ, GPIX, and GPV [81] and is the receptor for vWF but also for TSP-1, α-thrombin, and CD62P [80]. Different streptococcal species have been shown to bind to GPIbα via so-called glycosylated adhesins containing serine-rich repeats. This protein family was described in *Streptococcus*
*gordonii* (glycosylated streptococcal protein B, GspB) [82] and *Streptococcus*
*sanguinis* (serine-rich protein A, SrpA, and hemagglutinin salivary antigen, HSA) [83] and was shown to bind to sialic acid residues. In addition, vWF is used by bacteria for bridging to GPIbα as was shown for *S. aureus* surface protein A (SpA) and an uncharacterized *H.*
*pylori* surface protein [68,84].

TLRs are type 1 transmembrane proteins widely expressed on eukaryotic cells and best characterized on immune cells such as macrophages and dendritic cells. The ectodomain of TLRs contains leucine-rich β-sheets interacting with PAMPs and a Toll-interleukin-1 receptor domain for signal transduction [85]. Platelets express TLR1, TLR2, TLR4, TLR7, and TLR9, with TLR4 being the most abundant [86,87]. Next to the LPS-recognizing TLR4, TLR2 is also important in bacterial infections. TLR2 contains a glycosylated N-terminal ligand-binding domain with leucine-rich repeats [88] and recognizes bacterial lipoproteins [89]. Platelet TLR2 has been shown to be a target of *S. pneumoniae* and group B streptococci (GBS). Binding induces activation of the phosphoinositide-3 (PI3)-kinase pathway, finally leading to platelet activation and aggregation [90,91]. The interaction between pneumococci and platelet TLR2 induces further activation of integrin αIIbβ3 as well as release of dense granules [90]. Furthermore, TLR9 is also involved in bacteria-induced platelet aggregation. TLR9 recognizes cell-free bacterial DNA, which is increased in the blood of septic patients, leading to activation of coagulation [92].

In addition to the above-mentioned platelet receptors, the zinc-dependent metalloproteinase ADAM10 is a receptor for the *S. aureus* α-hemolysin (Hla), and binding leads to cleavage of GPVI [93]. Hla is a β-barrel pore-forming toxin, forming pores of 1–3 nm in diameter in the lipid bilayer of eukaryotic cells. These pores allow molecules up to a size of 4 kDa to pass through [94,95]. Hla is expressed by most *S. aureus* clinical isolates, and expression levels have been reported to correlate with virulence and disease severity [96,97]. Interactions of platelets with Hla have been reported to cause platelet activation and aggregation [98,99]. In addition, we recently demonstrated that Hla-mediated platelet activation is followed by loss of platelet function, leading to impaired thrombus formation and reduced stability of formed thrombi [100].

## 5. Platelets in *S. pneumoniae* Infections

*S. pneumoniae* exhibits the typical characteristics of Gram-positive bacteria. Pneumococci are enclosed by a flexible bilipid membrane, which is surrounded by a thick, multi-layered peptidoglycan sacculus composed of highly cross-linked glycan strands [101]. Besides peptidoglycan, teichoic acids are a general constituent of the Gram-positive cell wall [102]. Pneumococci possess rather complex, structurally unique, peptidoglycan-anchored wall teichoic (WTA) and membrane-anchored lipoteichoic acids (LTA), which are built up of identical repeating sugar units, highly decorated with phosphorylcholine [103,104]. Pneumococci shield themselves from the environment by a thick polysaccharide capsule enveloping the cell wall, whose composition is serotype-specific [105]. The capsule is the main virulence factor of pneumococci and protects the bacteria effectively from opsonization and phagocytosis by the host immune system [106]. Furthermore, four classes of proteins can be found on the surface of pneumococci, which can be classified by their mode of anchoring. The largest group with 37 predicted members is the group of lipoproteins, which are anchored to the bacterial cell membrane via an N-acyl diacylglycerol group [107,108]. Most of the lipoproteins are predicted to be part of ABC-transporters, which are essential for nutrient uptake and therefore directly involved in bacterial fitness [109]. Other lipoproteins have been shown to have essential functions in protein folding, cell wall biosynthesis, stress response, or pathogenicity [110,111,112,113]. The second group of pneumococcal surface proteins is the unique group of choline-binding proteins (CBPs). This group consists of 13–16 proteins (strain dependent), which contain N- or C-terminally a choline-binding domain composed of highly conserved choline-binding modules. CBPs are non-covalently bound to the phosphorylcholine moiety of the repeating units of WTA and LTA [114]. Well-characterized CBPs are the major autolysin LytA, which plays an important role in autolysis and virulence and the pneumococcal surface protein C (PspC), which is essential for pneumococcal colonization and pathogenesis [115,116,117,118,119]. The third class of pneumococcal surface proteins (about 13–19 members) contains a C-terminal cell wall-sorting signal beginning with a LPXTG amino acid motif [120,121]. The transpeptidase sortase A (SrtA) recognizes this motif; cleaves between the threonine and glycine residues; and anchors the protein to lipid II, which is subsequently incorporated into the peptidoglycan of the cell wall [122,123]. Known representatives from this group are, for example, the neuraminidase A (NanA) and the pneumococcal adhesion and virulence factor B (PavB), which were shown to be involved in pneumococcal adhesion and pathogenesis. The fourth group of pneumococcal surface proteins is the so-called moonlighting proteins, also known as non-classical surface proteins. This group includes enzymes that are actually ubiquitous intracellularly but can also be found on the surface of bacteria, where they play an additional role, often associated with virulence [124]. One example of such a protein is the pneumococcal enolase, which intracellularly converts 2-phosphoglycerate to phosphoenolpyruvate during glycolysis, but can also be found on the bacterial surface. Here, the enolase was shown to bind host plasminogen as well as the human complement inhibitor C4b-binding protein, leading to enhanced adherence to epithelial and endothelial cells and complement evasion [125,126].

### 5.1. Community Acquired Pneumonia (CAP)

*S. pneumoniae* is one of the leading causes of community-acquired pneumonia (CAP). CAP is a potentially life-threatening disease, with a mortality rate of up to 14% in hospitalized patients [127]. The highest risk for an infection with severe outcomes is in young children, the elderly, immunocompromised patients, and those with comorbidities [128]. During severe CAP, systemic platelet activation [129] and dropping platelet counts have been reported. The development of thrombocytopenia correlates with increased mortality [130,131]. As discussed in Section 5.4, *S. pneumoniae* directly and indirectly stimulates platelets, leading to activation and release of granule content, which is accompanied with the release of antimicrobial peptides (AMPs). However, although AMPs are released, *S. pneumoniae* is not affected by platelet releasates, but in turn destroys platelets themselves [132]. Neutrophils are one of the most important players in the progression of inflammation and also sepsis. In response to bacteria, neutrophils form NETs built up of neutrophil DNA, histones, and granular proteins such as defensins, thereby often leading to bacterial trapping and antimicrobial actions [133,134]. Neutrophil-derived defensins inhibit the synthesis of bacterial DNA, RNA, and proteins. In addition, lysozymes degrade the bacterial cell wall, and elastase cleaves bacterial surface virulence factors [135]. NETs and their extracellular histones mediate initiation and proceeding of platelet activation and coagulation, leading to a prothrombotic phenotype [136]. Pneumococci evade trapping in NETs by protective effects mediated by D-alanylation of LTA [137], blocking of NET binding via pneumococcal surface protein A [138], and by degrading the neutrophil DNA scaffold via endonuclease A [139]. Taken together, *S. pneumonia*e is able to evade the platelet induced immune response, and on the contrary triggers an inflammatory and coagulant phenotype of platelets during severe CAP.

### 5.2. Sepsis

Sepsis is a common complication of pneumococcal infections, with mortality rates of up to 30%. Sepsis leads to an activation of the coagulation cascade with consumption of circulating coagulation factors, platelets, and the generation of platelet–leukocyte complexes. As a result, patients develop thrombocytopenia and DIC, finally leading to hypoxic organ damage. Up to 50% of patients with severe sepsis develop DIC [140]. Several factors lead to DIC. First, coagulation is activated via different pathways [134,141,142]. Second, there is an increase of endothelial adhesion molecules, leading to platelet adhesion to endothelial cells and exposed subendothelial collagen. During these processes, platelet activation, aggregation, microthrombus formation, and finally vessel occlusion are triggered [99,143,144,145]. In addition, neutrophils are attracted and affected during sepsis. High numbers of circulating immature forms of neutrophils in the peripheral, impaired migration but also prolonged presence of NETS are commonly observed in sepsis [146,147]. NET formation indirectly triggers tissue factor release of endothelial cells, which is then captured in NETs, further triggering coagulation [144,148]. Another factor is the massive amount of released reactive oxygen species (ROS). ROS favor vasoconstriction and have an activating effect on platelets [149]. Next to aggregation and thrombus formation, platelets also play a role in inflammatory processes, which are associated with sepsis. Platelet-derived soluble CD40L is increased in the circulation of septic patients [150,151,152] and plays a central role in activation and recruitment of neutrophils by activation of the β2 integrin Mac-1 of neutrophils and indirectly via macrophage inflammatory protein-2 and subsequent CXCR2 signaling [153,154]. Under septic conditions, activation of CXCL4 is also increased. This results in platelet degranulation, release of proinflammatory factors, and stimulation of the coagulation cascade [155]. Taken together, bacterially induced microthrombus and NET formation can cause uncontrolled coagulation and inflammation, leading to thrombocytopenia and DIC during sepsis.

### 5.3. Infective Endocarditis

Development of infective endocarditis is a complication of bacteremia caused by *S. pneumoniae*. In the pre-antibiotic era, *S. pneumoniae* caused about 15% of IE cases. Upon usage of penicillin or cephalosporine, the prevalence dropped to approximately 3% in the 1990s, and currently, prevalence data are missing [156,157]. IE caused by pneumococci mainly affects the aortic valves, and patients often also suffer from acute pneumonia; a common complication of pneumococcal IE is embolism [158]. Before bacteria can colonize the valves and cause IE, the surface structure of the endothelial valves has to be altered, as observed after previously occurred endocarditis or valve replacement [159]. On these structural changes, fibrin and platelets adhere, forming the so called non-bacterial thrombotic vegetation (NBTV) [160,161]. These NBTV serve as a niche for colonization in transient bacteremia or in the case of oral streptococci, after dental treatment [162,163,164], leading to further platelet aggregation on the surface and fibrin deposition [160]. By further acquisition of fibrin and platelets, bacteria are shielded from the immune system, allowing bacteria to reach very high densities [24,165].

### 5.4. Pneumococcal Interactions with Platelets

In contrast to *S. aureus*, little is known about the interaction between *S. pneumonia*e and platelets. The first studies describing platelet activation due to pneumococci were published in the 1970s. These studies showed that some pneumococcal serotypes induced platelet activation and aggregation in vitro, whereas other serotypes had no effect on platelet activation [166,167]. Later, platelet aggregation was shown to be induced only in interactions with encapsulated strains via an interaction with TLR2, but not with nonencapsulated strains [90]. However, other studies reported contradictory results. De Stoppelaar and colleagues did not observe platelet aggregation in encapsulated strains [168]. In addition, a TLR2-independent platelet degranulation was observed, which could be confirmed in mice with knockouts for several TLRs [168]. However, recent studies, including our own, demonstrated a direct binding of pneumococci to platelets. One study showed aggregate formation between platelets and pneumococci. This aggregate formation was dependent on the presence of soluble fibrin and the presence of TSP-1 derived from activated platelets [169]. The pneumococcal adhesins PavB and PspC are hypothesized to bind to platelet GPII/bIII via bridging of TSP-1 [170,171].

One of the major virulence factors of *S. pneumoniae* is pneumolysin (Ply), a cholesterol-dependent cytolysin that oligomerizes into the membrane after binding to the target cell, leading to pore formation [172]. Ply has been shown earlier to activate and aggregate platelets depending on Ca ^2+^-influx through pneumolysin pores [173,174]. However, our own data demonstrate platelet killing by pneumolysin (Figure 4) [47].

In accordance with other studies, we also observed highly increased P-selectin signals in flow cytometry, suggesting massive platelet activation. However, instead, as shown by confocal imaging, only intracellular stores of P-selectin were stained due to antibodies diffusing through the pneumolysin pores. In addition, the increase in light transmission of the platelet suspension after incubation with pneumolysin is due to cell lysis instead of aggregation. Platelet lysis occurred immediately after addition of pneumolysin, even at very low concentrations. A loss of platelet function was also observed in whole blood experiments [47]. Nevertheless, we also observed increased P-selectin signals in lysates of *ply* knockout strains, which appear independent of pneumococcal-derived H_2_O_2_, since P-selectin levels were similar in a *ply*-mutant and a Δ*spxB*Δ*ply* double mutant (Figure 5B). Therefore, we hypothesized that other surface-associated proteins of *S. pneumoniae* trigger platelet activation. A screening of our library of pneumococcal surface proteins (Table 1) in activation assays with washed platelets revealed candidate proteins, which directly activate platelets. Among them were SP_0899, a lipoprotein of thus far unknown function, and CbpL, a choline-binding protein and putative adhesion contributing to colonization [175]. In addition, AliB and SP_1833 also induced at least a slightly increased P-selectin staining. AliB is a lipoprotein and functions as a substrate-binding protein for oligopeptides [176], and SP_1833 (PfbA) is a sortase-anchored protein with plasmin- and fibronectin-binding capacity [177] (Figure 3A). All tested pneumococcal lipoproteins were heterologously expressed without the lipid moiety. The naturally occurring lipidation of lipoproteins has been shown to trigger TLR2-dependent signaling in leukocytes [178,179]. Therefore, lipidated proteins were tested in comparison to their non-lipidated forms [179], but no activation leading to P-selectin surface staining was detected. Nevertheless, it is noteworthy to know that TLR expression levels in resting platelets is low, but they become upregulated, and levels increase on the platelet surface upon activation [87,180]. Since proteins with platelet activation potential were identified in all groups of pneumococcal surface proteins, mutants were applied in activation assays lacking pneumolysin or whole groups of surface proteins [107,177]. This was achieved by the deletion of genes encoding essential enzymes involved in the anchoring of proteins (prolipoprotein diacylglyceryl transferase Lgt for lipoproteins and SrtA for sortase-anchored proteins) to the bacterial surface. To remove CBPs from the surface of pneumococci, a *ply* mutant was treated with choline chloride. During severe invasive infections, antibiotic treatment autolysis as well as antibiotic-induced lysis occurs in the bloodstream. Therefore, lysates generated from these deletion or choline chloride-treated strains were also tested for their platelet activation potential. Independent of the genetic background, all lysates induced increased P-selectin signals at similar levels (Figure 1B), leading to the assumption that another factor, e.g., components of the cell wall could be responsible for platelet activation. However, isolated and structurally defined protein-free pneumococcal lipoteichoic acids as well as wall teichoic acids [181] had no impact on platelet P-selectin surface expression (Table 1). Taken together, besides the identified surface proteins, another thus far unknown factor might be able to induce direct platelet activation. Nevertheless, the lytic effect of pneumolysin on platelets, even at very low concentrations, probably overshoots any other effect of pneumococcal proteins on platelets in invasive infections (Figure 1 and Figure 4).

## 6. Relevance of Findings for Disease

Thus far, there are only a few studies focusing on platelet activation during pneumococcal infections. One study showed that the expression of *pblB*, a phage-derived gene, was associated with increased platelet activation and mortality in hospitalized patients suffering from CAP caused by *S. pneumoniae* [129]. Another study showed increased platelet activation and platelet hyperreactivity in a porcine model of invasive *S. pneumoniae* infections [182]. In a follow-up in vitro study, the authors demonstrated that desialylation of platelets by the pneumococcal neuraminidase A (NanA) results in hyperreactivity of platelets to ADP stimulation [183].

One approach to interfere with platelet dysfunction in pneumococcal infections is directly targeting and neutralizing the pore forming pneumolysin with antibodies. Promising candidates are the pharmaceutically available IgG preparation IVIG (Privigen, 98% IgG) or the mixed immunoglobulin preparation trimodulin (21% IgA, 23% IgM, 56% IgG). Both immunoglobulin preparations contain antibodies against pneumolysin and have been shown to efficiently inhibit platelet damage in vitro, as shown by rescued platelet function and viability even in the presence of high pneumolysin concentrations (Figure 4) [47,184]. The presence of IgM and IgA in trimodulin had no beneficial effect for pneumolysin neutralization. In fact, neutralization efficiency was dependent on IgG content in the immunoglobulin preparation [184]. In a phase 2 clinical trial (CIGMA study), patients with severe community-acquired pneumonia were treated with trimodulin in addition to standard care. In the trimodulin group, patients had higher platelet counts and a nominally lower mortality compared to the placebo group [47]. However, patients included in the study were only 160, and larger clinical trials are necessary to confirm this observation.

A major cause for thrombocytopenia is coagulopathy conditions, observed in 80% of septic patients, with DIC being the most severe form [185]. Therefore, treatment of coagulopathies is highly needed to reduce mortality rates and coagulation-associated tissue damage. The focus of ongoing research lies on inhibition of coagulation or the use of anti-platelet drugs. Anti-platelet therapy during sepsis or ARDS is not part of standard care, but several studies conclude this as a promising approach to reduce disease severity [186,187,188,189]. However, the benefit of anti-platelet drugs such as, e.g., acetylsalicylic acid during sepsis is under debate [189,190], and more stringent clinical studies are needed. The same accounts for inhibition of coagulation. Although some clinical studies conclude damped disease severity upon usage of, e.g., antithrombin, its benefit is controversial, and more research on this topic is needed [191,192].

## 7. Conclusions

In this review, we highlight interactions between platelets and *S. pneumoniae* or *S. aureus* in vitro and in disease. Multiple factors of both bacteria result in direct or indirect platelet activation and aggregation. However, the predominant effect seems to be exerted by the pore-forming toxins pneumolysin (pneumococci) and alpha-hemolysin (Hla, *S. aureus*). Whereas Hla first activates and later on lyses platelets, pneumolysin directly lyses platelets. The pathomechanisms of these toxins and their impact on platelet function are of high clinical importance because thrombocytopenia and DIC are typical complications in severe invasive infections caused by these pathogens. Future clinical studies targeting either bacterial components such as pneumolysin and Hla and their receptors or systemic coagulation as seen in dependence of Hla are highly needed to improve clinical outcomes.

## Figures and Tables

**Figure 1 cells-11-01121-f001:**
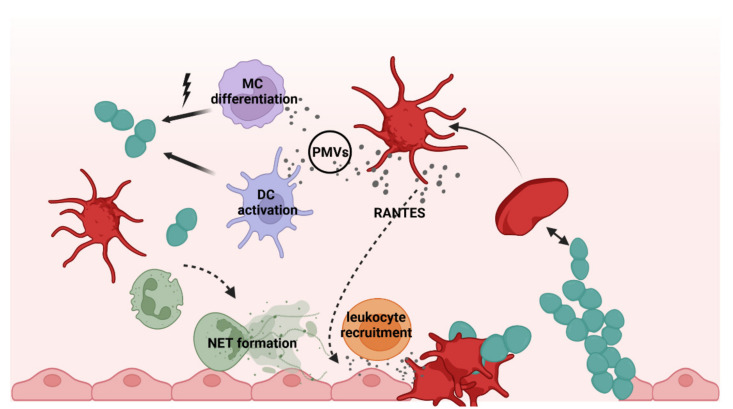
Scheme illustrating different platelet functions in the immune response. Platelets sense and bind invading bacteria and injured endothelium, resulting in platelet activation. Upon activation, platelets release chemokines and cytokines such as RANTES, triggering leukocyte recruitment and PMVs acting on gene expression of monocytes and monocyte (MC)-derived cells such as dendritic cells (DC). In addition, neutrophils are attracted, and NET formation occurs at the site of infection via platelet-dependent mechanisms. Created with BioRender.com (accessed on 21 March 2022).

**Figure 2 cells-11-01121-f002:**
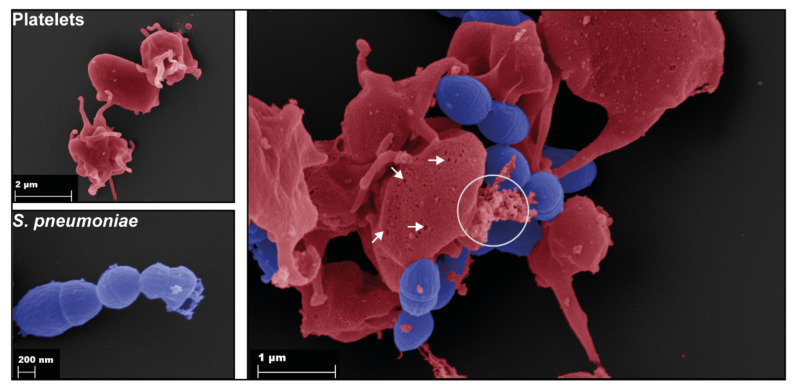
Binding of *S. pneumoniae* (blue) to platelets (red). Scanning electron microscopy of single platelets (upper left), single pneumococci (bottom left), and platelets incubated with the pneumococcal TIGR4 strain for 1 h (right). The right image shows binding of pneumococci to platelets. In addition, pneumolysin pores are formed in platelet membranes (arrows), and released granule content is visible (circle).

**Figure 3 cells-11-01121-f003:**
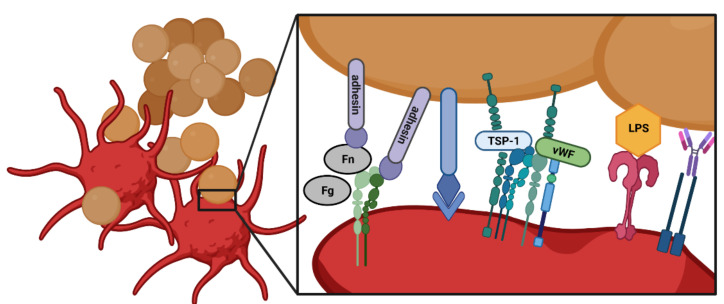
Binding of bacteria to platelets occurs either directly or indirectly. Bacterial adhesins with specific repeating units can utilize ECM proteins such as fibronectin (Fn), fibrinogen (Fg), TSP-1, or vWF as molecular bridges to bind to, e.g., integrin αIIβ3 or other complexes of glycoproteins. Furthermore, some bacterial factors can directly bind to integrins, TLRs, or other platelet surface proteins. Bacteria already covered by IgGs are recognized by FcγRIIa. Created with BioRender.com (accessed on 21 March 2022).

**Figure 4 cells-11-01121-f004:**
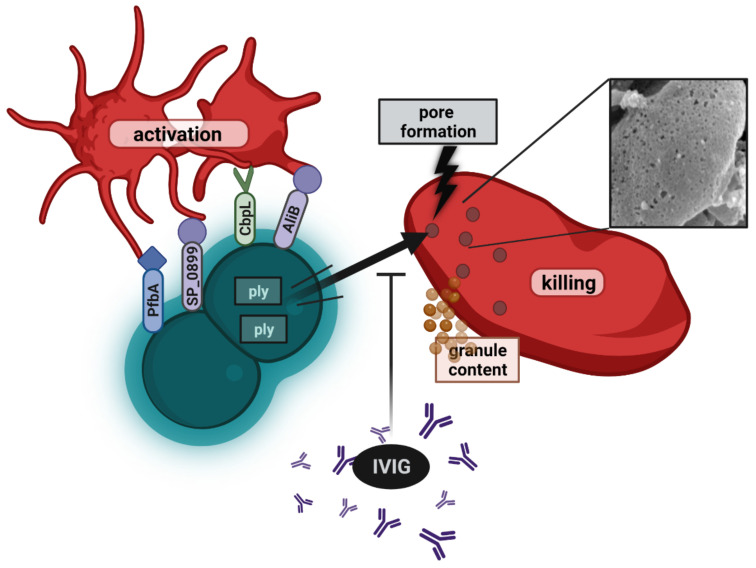
Scheme of different interactions between platelets and pneumococci. Individual pneumococcal surface proteins can induce direct activation of platelets. On the other hand, the intracellular pneumolysin (Ply) kills platelets by extensive pore formation in platelet membranes, as shown by the SEM image of a Ply-treated platelet on the right side. Pneumolysin is released in the circulation upon autolysis of pneumococci, and its action on platelets can be neutralized by the addition of pharmaceutical IgG preparations. Created with BioRender.com (accessed on 21 March 2022).

**Figure 5 cells-11-01121-f005:**
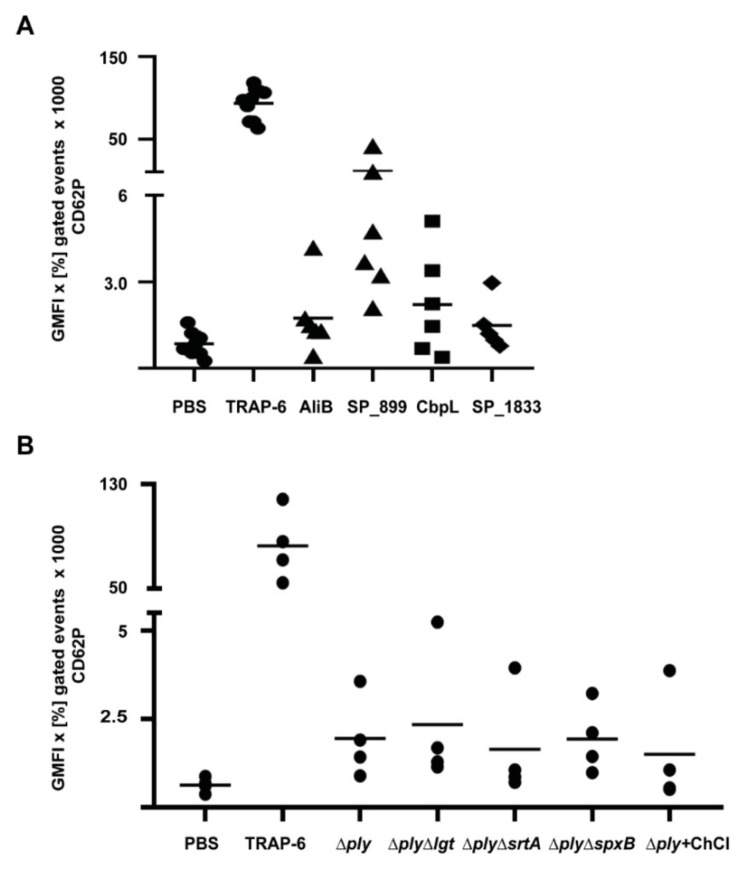
Individual pneumococcal proteins and pneumococcal lysates directly activate human platelets. Washed platelets of a defined set of donors were incubated with different concentrations of pneumococcal proteins (**A**) (Table 1) for 30 min or pneumococcal lysates (**B**) with the indicated genetic backgrounds for 60 min at 37 °C. CD62P was used as an activation marker and was detected by flow cytometry, using a PE-Cy5-labelled P-selectin antibody. PBS was used as negative control, and 20 µM TRAP-6 was used as a positive control. The data are presented as geometric mean of fluorescence intensity (GMFI) of positive gated events multiplied with the percentage of positive gated events in the dot plots.

**Table 1 cells-11-01121-t001:** List of pneumococcal proteins and cell wall components, which were tested to activate platelets and led to CD62P expression. * The last column provides the highest tested concentration of each protein in the platelet activation assay. The molarities were chosen on the basis of previous publications determining platelet activating potential of bacterial proteins [24].

Protein Class	No.	Protein Name SP Number	Function	Activation of Washed Platelets	Protein Concentration (µM) *
Lipoproteins	1	AdcAII (SP_1002)	substrate-binding protein of ABC transporter for zinc(II) ions	-	4
	2	AliB (SP_1527)	substrate-binding protein of ABC transporter for oligopeptides	+	2
	3	AliC	substrate-binding protein of ABC transporter for oligopeptides	-	4
	4	AliD	substrate-binding protein of ABC transporter for oligopeptides	-	4
	5	AmiA (SP_1891)	substrate-binding protein of ABC transporter for oligopeptides	-	4
	6	DacB (SP_0629)	L,D-carboxypeptidase, peptidoglycan turnover	-	4
	7	Lipidated DacB	L,D-carboxypeptidase, peptidoglycan turnover	-	4
	8	Etrx1 (SP_0659)	extracellular thioredoxin protein 1	-	4
	9	Etrx2 (SP_1000)	extracellular thioredoxin protein 2	-	4
	10	MetQ (SP_0149)	substrate-binding protein of ABC transporter for methionine	-	4
	11	Lipidated MetQ	substrate-binding protein of ABC transporter for methionine	-	4
	12	PccL (SP_0198)	transport of small hydrophobic molecules such as siderophores	-	4
	13	PiaA (SP_1032)	substrate-binding protein of ABC transporter for iron	-	4
	14	PnrA (SP_0845)	substrate-binding protein of ABC transporter for nucleosides	-	4
	15	PpmA (SP_0981)	proteinase maturation protein, peptidyl-prolyl isomerase	-	4
	16	PsaA (SP_1650)	substrate-binding protein of ABC transporter for manganese	-	4
	17	SlrA (SP_0771)	streptococcal lipoprotein rotamase, peptidyl-prolyl isomerase	-	4
	18	GshT (SP_0148)	substrate-binding protein of ABC transporter for glutathione	-	4
	19	SP_0191	unknown function	-	4
	20	SP_0899	unknown function	+++	2/4
	21	FusA (SP_1796)	substrate-binding protein of ABC transporter for fructo-oligosaccharides	-	2
	22	RafE (SP_1897)	substrate-binding protein of ABC transporter for multiple sugars	-	4
	23	PstS (SP_2084)	substrate-binding protein of ABC transporter for phosphate ions	-	-
	24	SP_1690	substrate-binding protein of ABC transporter	-	4
	25	MalX (SP_2108)	substrate-binding protein of ABC transporter for maltose/maltodextrin	-	4
	26	SatA (SP_1683)	substrate-binding protein of ABC transporter for sialic acid	-	4
CBPs	27	CbpC (SP_0377)	regulatory function for autolysis by inhibiting autolysin LytC	-	4
	28	CbpF (SP_0391)	putative adhesin	-	4
	29	CbpL (SP_0667)	putative adhesin	++	4
	30	Chimeric (PspA+PspC)	fusion of N-terminal domains of PspA and PspC	-	4
	31	PcpA (SP_2136)	adhesin	-	2
	32	PspA_QP2 (SP_0117)	virulence factor, binds lactoferrinand inhibits complement activation	-	4
	33	PspC_SH2 (SP_2190)	adhesion, IgA inactivation, major factor H–binding protein	-	4
Sortase– anchored proteins	34	PfbA (SP_1833)	plasmin- and fibronectin-binding protein	+	2
	35	PitB (spt_1059)	pilin of pneumococcal pilus-2, adhesin	-	2
	36	PsrP (SP_1772)	adhesion, biofilm formation	-	4
	37	RrgB (SP_0463)	pilus-1 anchorage protein	-	4
	38	RrgC (SP_0464)	pilus-1 backbone protein, pilin	-	4
	39	SP_1992	adhesin architecture, bind to collagen and lactoferrin in vitro	-	4
Cell wall components	40	lipoteichoic acids		-	4
	41	wall teichoic acids		-	40 µg/mL

## Data Availability

Data is contained within the article.

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
