# Peer review of "Platelets, Bacterial Adhesins and the Pneumococcus"

_cells, 2022, doi:10.3390/cells11071121_

Round 1

Reviewer 1 Report

This manuscript is an overview of the interplay between platelets and bacteria via adhesin-receptor interactions and platelet-activating bacterial proteins with a focus on S. Aureus and S. Pneumonia. This review is well documented and clear but need to be implemented by figures for the reader to have a global picture of the certain aspects.

  1. Please add figures that give an overview of platelets as immune cells in infections (for paragraph 2) and of platelet receptors in bacterial infections and bacterial adhesins (for paragraph 4).
  2. The following review need to be cited in 5.2 part: Platelets Are Critical Key Players in Sepsis. Vardon-Bounes F, Ruiz S, Gratacap MP, Garcia C, Payrastre B, Minville V. Int J Mol Sci. 2019 Jul 16;20(14):3494. doi: 10.3390/ijms20143494.
  3. Please rephrase the abstract lines 15-16 and 17
  4. Spelling errors: Line 151: Please change GP II2/IIa and GPIIa/IIb as it does not exist; Line 202: there is a misspelling on phosphoinositide; please go through the text as there are some spelling errors.

Author Response

This manuscript is an overview of the interplay between platelets and bacteria via adhesin-receptor interactions and platelet-activating bacterial proteins with a focus on S. aureus and S. pneumoniae. This review is well documented and clear but need to be implemented by figures for the reader to have a global picture of the certain aspects.

 Please add figures that give an overview of platelets as immune cells in infections (for paragraph 2) and of platelet receptors in bacterial infections and bacterial adhesins (for paragraph 4).

The following review need to be cited in 5.2 part: Platelets Are Critical Key Players in Sepsis. Vardon-Bounes F, Ruiz S, Gratacap MP, Garcia C, Payrastre B, Minville V. Int J Mol Sci. 2019 Jul 16;20(14):3494. doi: 10.3390/ijms20143494.

Please rephrase the abstract lines 15-16 and 17

Spelling errors: Line 151: Please change GP II2/IIa and GPIIa/IIb as it does not exist; Line 202: there is a misspelling on phosphoinositide; please go through the text as there are some spelling errors.

Response:

We thank this reviewer for the overall positive evaluation of our manuscript.

We included all points raised by the reviewers in the revised version of our manuscript and also added two new Figures for better clarity.

Reviewer 2 Report

In this review manuscript by Jahn and colleagues, the functions of platelets in host defense and the interactions of platelets with microorganisms (mainly gram positive bacteria) are described.

The work has the potential to become a good reference for readers interested in platelet activities beyond those in hemostasis, but the manuscript in its present form is far from exploiting this potential.

The manuscript suffers from a lack of structure and an unclear distinction of main and side issues (focus), leading it to be overinflated. Perhaps this is due to a suboptimally coordinated contribution of the different authors. This reviewer strongly recommends that the final revised manuscript is thoroughly redacted by the PI.

In addition, many references are ancient and appear not optimally selected. Also here, a more careful and economic selection is recommended.

Specific comments:

  • chemokines should be listed primarily with their systematic names (CCL5, CXCL4 etc.)
  • integrin GPIIb/IIIa should be consistently written as αIIbβ3 (also see 2 misspellings in line 151)
  • Figure 1 would benefit from arrows at points of interest. 
  • Lines 34-39 and 48-51 appear to contain repeated information
  • Line 72: the platelet as a ("the main"!) source of IL-1b is highly controversial and the studies by Brown required addition of soluble CD14 to become measurable. I recommend reformulating this section.
  • Line 85: "shedding" of platelets is somewhat inaccurate wording. Is there any clinical evidence for this observation (ref. 24)?
  • Section starting with line 86: This should be expanded and should contain the right references. For example, platelet release of CCL5 and leukocyte recruitment (Pubmed ID: 11282909). The interaction of P-selectin and monocytes (8617886, 10515876). Reference 26, line 95 appears inappropriate.
  • Section starting on line 99: Please also discuss the work (15890969, 20110505, 21956548, 26333874, 28717419). Please use the modern term platelet "microvesicles"
  • Line 109: there are four classes of PMPs (this appears wrong)
  • Line 125: perhaps it is worth mentioning which toxins are released and what they do.
  • Line 128: it is well worth discussing the excellent studies by Gärtner (PubmedID: 29195076) and Verschoor (22037602).
  • Section 4 needs to be structured more stringently. Either from the perspective of the platelet receptor (as target for adhesins) or from the perspective of the adhesins. Now it appears to move back and forth. 
  • Line 155: This should be introduced earlier.
  • Line 160: this should be rephrased and connected better to the subsequent section (e.g. "Some bacteria express surface proteins containing domains that are rich in serine-aspartate repeats. This is highlighted by a family of S. aureus surface components...")
  • Line 212: what correlates? Disease severity or virulence?
  • Please consider shortening and streamlining section 5.
  • Lines 255-258: should appear earlier
  • Line 269: the description of NETs (line 300) should appear here.
  • Line 284 and forth: Please briefly describe the function of plasminogen-activator inhibitor 1. Activated protein C is an inhibitor of coagulation OR an inactivator of coagulation factors
  • Line 290: tissue factor (not factors). To the reviewer's best knowledge, monocytes are rather a source. Neutrophils release NETs, indirectly leading to TF expression on the endothelial cells (e.g. 29976772).
  • Line 296: please check "macrophage-1 antigen"
  • Line 317: How does the mentioned alteration of valve surface happen? (isn't there a newer reference for 141?)
  • Line 334: It is absolutely no disgrace mentioning that the studies discussed are your own.
  • Table 1: What information is provided by the concentrations used? Perhaps the EC50 is worth mentioning rather than max. concentration tested (for the none-inducers: n.d.).
  • Line 445: For ARDS, the work by Hidalgo is worth mentioning: 19305412
  • Section starting from line 458: this lacks focus. What is the reason for discussing antiplatelet therapy? Is this applied e.g. in sepsis, pneumonia or endocarditis?
  • Section 6 appears to die out somewhat at the end. Perhaps the authors can write a section with a clear outlook for their findings and for those of others.

Author Response

In this review manuscript by Jahn and colleagues, the functions of platelets in host defense and the interactions of platelets with microorganisms (mainly gram positive bacteria) are described.

The work has the potential to become a good reference for readers interested in platelet activities beyond those in hemostasis, but the manuscript in its present form is far from exploiting this potential.

The manuscript suffers from a lack of structure and an unclear distinction of main and side issues (focus), leading it to be overinflated. Perhaps this is due to a suboptimally coordinated contribution of the different authors. This reviewer strongly recommends that the final revised manuscript is thoroughly redacted by the PI.

In addition, many references are ancient and appear not optimally selected. Also here, a more careful and economic selection is recommended.

Response:

We are grateful for this detailed review and the reviewers suggestions. We have carefully and significantly revised our review and considered all the points raised by the reviewer.

See below the changes that have been done according to the suggestions. A labelled revised version is  submitted only for the reviewers to ease the review process.

Specific comments:

  • chemokines should be listed primarily with their systematic names (CCL5, CXCL4 etc.)
  • integrin GPIIb/IIIa should be consistently written as αIIbβ3 (also see 2 misspellings in line 151)
  • Figure 1 would benefit from arrows at points of interest. 
  • Lines 34-39 and 48-51 appear to contain repeated information

Response:

all points have been improved.

In the revised version we mentioned this now as follow:

“Besides the above mentioned proteins they also contain proteins acting as chemokines and cytokines like e.g. RANTES (Regulated upon Activation, Normal T Cell Expressed and Presumably Secreted), or stromal cell-derived factor 1 (SDF-1) and recruit and stimulate other cells of the immune system or induce endothelial inflammation16.”

  • Line 72: the platelet as a ("the main"!) source of IL-1b is highly controversial and the studies by Brown required addition of soluble CD14 to become measurable. I recommend reformulating this section.

Response:

We have rephrased the sentence;

New version:

„In addition, next to inflammatory and immune cells also platelets produce interleukin 1-β (IL-1β),“ which is …

  • Line 85: "shedding" of platelets is somewhat inaccurate wording. Is there any clinical evidence for this observation (ref. 24)?

Response:

We have rephrased the sentence;

New version:

 “In addition, TLR2 stimulation of MKs is followed by increased maturation of megakaryocytes  and elevated protein content, suggesting effects on platelet count, platelet function and inflammation 26

  • Section starting with line 86: This should be expanded and should contain the right references. For example, platelet release of CCL5 and leukocyte recruitment (Pubmed ID: 11282909). The interaction of P-selectin and monocytes (8617886, 10515876). Reference 26, line 95 appears inappropriate changed

Response:

We have improved and expanded the section;

New version (line 109-115)

 “Platelet derived chemokines like RANTES trigger activation and arrest of T-cells at sites of infection. Upon release, RANTES is immobilized on the endothelium and triggers arrest of monocytes and monocyte-derived cells at the endothelium as well as on adherent platelets in a shear resistant manner28. Adhesion of rolling monocytes on activated platelets but also on the exposed endothelium is P-selectin dependent and this interaction further triggers expression and secretion of cytokines like TNF-α29-31.”

  • Section starting on line 99: Please also discuss the work (15890969, 20110505, 21956548, 26333874, 28717419). Please use the modern term platelet "microvesicles"

Response:

We thank the reviewer for this excellent suggestions. We have therefore rephrased the paragraph;

New version: (line 121-138)

“Another immune cell function of platelets is displayed by the release of microvesicles. The so called platelet microvesicles (PMVs) are lipid membrane vesicles with a size of 0.1-1.0 µm mediators of cell-cell communication. Elevated numbers of PMVs in the circulation are associated with inflammation and diseases like arthritis36, development of an acute coronary artery syndrome and stroke 37-38. PMVs carry adhesion molecules (CD62P, RANTES), which facilitate monocyte arrest at sites of deposed PMVs (vessel walls) and additional recruitment of activated platelets39. They contain up to 250 different microRNAs (miRNAs, 20-22 nucleotides long). These miRNAs and also other stored molecules can be transferred to other cells types like immune cells, vascular cells but also smooth muscle cells thereby changing their gene expression40-41. Changes in gene expression include phenotypic switches of monocytes and monocyte-derives cell lines like macrophages towards a phagocytic phenotype42.Further, not just PMVs released upon platelet activation but also upon apoptosis of platelets have immunomodulatory effects, as shown by differentiation of monocytes into phagocytes43. In bacterial infections platelets can contribute to clearance of bacteria via release of antimicrobial peptides present in PMVs or stimulation of immune cells via release of immunomodulatory molecules.  PMVs can be distinguished in kinocidins, defensins, thymosins and derivatives of antimicrobial peptides, which act against Staphylococcus aureus (S. aureus) and Candida albicans 44-45.”

  • Line 109: there are four classes of PMPs (this appears wrong)

Response:

We have rephrased the sentence; line 136-138

New version:

 “PMVs can be distinguished in kinocidins, defensins, thymosins and derivatives of antimicrobial peptides, which act against Staphylococcus aureus (S. aureus) and Candida albicans 44-45

  • Line 125: perhaps it is worth mentioning which toxins are released and what they do.

Response:

We have rephrased the sentence; line 151-154

New version:

“To these belong e.g., Helicobacter pylori mediated immune thrombocytopenia driven by autoantibodies destroying platelets 51,52 or the shiga toxin-producing E. coli caused hemolytic uremic syndrome (HUS)52-53 which is due to amplification of the platelet activation process.”

  • Line 128: it is well worth discussing the excellent studies by Gärtner (PubmedID: 29195076) and Verschoor (22037602).

Response:

We have included this excellent study and apologize for having not considered this important study in our first version.

Line 165-169

“ Other studies demonstrated kind of a searching and shuttling of invading bacteria by platelets. Adherent platelets can migrate over their substrate collecting all substrate-bound material including bacteria, resulting in boosted activity of phagocytes61. In addition platelets were shown to deliver the intracellular bacterium Listeria monocytogenes to dendritic cells62.”

  • Section 4 needs to be structured more stringently. Either from the perspective of the platelet receptor (as target for adhesins) or from the perspective of the adhesins. Now it appears to move back and forth. 
  • Line 155: This should be introduced earlier.

Response:

We have rephrased this section (line 173-251) and have included “line 155” earlier in the introduction.

  • Line 160: this should be rephrased and connected better to the subsequent section (e.g. "Some bacteria express surface proteins containing domains that are rich in serine-aspartate repeats. This is highlighted by a family of S. aureus surface components...")
  • Line 212: what correlates? Disease severity or virulence?

Response:

This has been rephrased (now line 197-198) and “line 212” improved (now line 247)

Line 247: “Hla is expressed by most S. aureus clinical isolates and expression levels have been reported to correlate with virulence and disease severity 97-98.”

  • Please consider shortening and streamlining section 5.

Response:

We highly appreciate this comment by the reviewer and now focused this section more stringently, cut out side information and omitted the errors.

  • Lines 255-258: should appear earlier
  • Line 269: the description of NETs (line 300) should appear here.
  • Line 284 and forth: Please briefly describe the function of plasminogen-activator inhibitor 1. Activated protein C is an inhibitor of coagulation OR an inactivator of coagulation factors

Response:

We have improved these paragraphs and the rephrased paragraphs can be found now in section 5.1

We reconsidered “line 284 and forth” and decided to delete this part, because it fits not perfectly to the scope of the review

  • Line 290: tissue factor (not factors). To the reviewer's best knowledge, monocytes are rather a source. Neutrophils release NETs, indirectly leading to TF expression on the endothelial cells (e.g. 29976772).

Response:

We have improved this statement. (line 326-331)

New version:

“In addition, neutrophils are attracted and affected during sepsis. High numbers of circulating immature forms of neutrophils in the peripheral, impaired migration but also prolonged presence of NETS are commonly observed in sepsis148-149. NET formation indirectly triggers tissue factor release of endothelial cells, which is then captured in NETs further trigger coagulation150-151

  • Line 296: please check "macrophage-1 antigen"

Response:

This has been rephrased and corrected

New version:

“Platelet-derived soluble CD40L is increased in the circulation of septic patients153-155 and plays a central role in activation and recruitment of neutrophils by activation of the β2 integrin Mac-1 of neutrophils and indirect via macrophage inflammatory protein-2 and subsequent CXCR2 signalling156-157

  • Line 317: How does the mentioned alteration of valve surface happen? (isn't there a newer reference for 141?)

Response:

This has been rephrased and corrected (line 348-352)

New version:

 “Before bacteria can colonize the valves and cause IE, the surface structure of the endothelial valves has to be altered, as seen after previously occurred endocarditis or valve replacement162. On these structural changes fibrin and platelets adhere, forming the so called non-bacterial thrombotic vegetation (NBTV)163-164.”

New references (163 and 164) have been added: https://www.ncbi.nlm.nih.gov/pmc/articles/PMC8227130/, https://pubmed.ncbi.nlm.nih.gov/16953053/

  • Line 334: It is absolutely no disgrace mentioning that the studies discussed are your own.
  • Table 1: What information is provided by the concentrations used? Perhaps the EC50 is worth mentioning rather than max. concentration tested (for the none-inducers: n.d.).
  • Line 445: For ARDS, the work by Hidalgo is worth mentioning: 19305412

Response:

The concentrations used are now explained in the legend of the table. In fact, two concentrations are always used for the tests, identical to the study by Binsker et al., 2018 (reference 24: https://pubmed.ncbi.nlm.nih.gov/29554697/)

  • Section starting from line 458: this lacks focus. What is the reason for discussing antiplatelet therapy? Is this applied e.g. in sepsis, pneumonia or endocarditis?

Response:

We thank the reviewer for pointing to a missing focus. We have discussed this issue with a clinician and platelet expert and have decided to delete this paragraph. The use of anti-platelet therapy and inhibition of coagulation is under debate and part on ongoing research. We focus now on potential therapeutic approaches based on (our own) data discussed in the review.

  • Section 6 appears to die out somewhat at the end. Perhaps the authors can write a section with a clear outlook for their findings and for those of others.

Response:

We have added a new paragraph with our conclusions

Round 2

Reviewer 2 Report

The manuscript has been greatly improved and I recommend acceptance.